# OpenReview forum: "Understanding Catastrophic Forgetting in Language Models via Implicit Inference"
_ICLR.cc/2024/Conference — ICLR 2024 poster_

### Official Review · Reviewer_MyAe · 2023-10-15

**Soundness:** 2 fair
**Presentation:** 3 good
**Contribution:** 2 fair
**Rating:** 3
**Confidence:** 5

**Summary:**

The paper proposes a theory of "implicit task inference" to model the behavior of language models in finetuning, which offers an explanation of the phenomenon of catastrophic forgetting. The paper then proposes a "conjugate prompting" technique to recover suppressed pretraining capabilities by making a prompt appear farther from the fine-tuning distribution. The paper uses a synthetic linear regression experiment to validate the proposed theory, and uses a synthetic ICL dataset and adversarial examples to examine the effectiveness of the proposed conjugate prompting technique on language models. Results show that conjugate prompting partially recovers lost ICL performance and harmful content generation of finetuned language models.

**Strengths:**

* The study introduces a fresh perspective on the issue of catastrophic forgetting through the lens of inferencing task identity. The theoretical formulation is neat and reasonable, showing that adaptation of task probabilities in finetuning is a potential explanation of catastrophic forgetting in language models.
* The proposed conjugate prompting technique is easy to understand and apply. As a simple translation, it is universally applicable to most tasks.
* The concepts and methods are presented clearly, allowing for easy understanding of the proposed methodology. The graphical visualizations also significantly support the clarity of the presentation.

**Weaknesses:**

* The paper's main weakness is the main assumption being restrictive and artificial, therefore failing to reflect the behavior of language models in most realistic settings. The main assumption of the paper (Equation 7) is that the input space of different tasks overlaps, so the model has to do probabilistic inference on the task identity. Both the proposed synthetic regression task and the ICL examples have this overlap. The ICL examples are purposefully designed to make it confusing by including two different instructions. Such examples are adversarial in nature and unlikely to exist in real-world applications of language models, where the task is usually clearly specified in the prompt. For example, the input of the summarization task "Please summarize the following passage: [text]" is very unlikely to be confused with the input of a translation task "Please translate the following passage to English: [text]" given a random passage in place of [text].

  It can also be argued that when the input space of different tasks overlaps, the problem itself is ill-posed as there is not a single correct answer for inputs from the intersection. The failure of finetuned models thus may come from the ambiguity of the task itself rather than from catastrophic forgetting.

* The proposed theory does not seem to explain the main part of catastrophic forgetting. Per the author's theory, if the tasks' input spaces don't overlap, the model does not need to adjust the task probability $g$ (as it would be already close to 1 for a good enough language model), and catastrophic forgetting would be much less if $g$ does not need to be modified. However, catastrophic forgetting is quite ubiquitous in practice for non-overlapping tasks and regardless of how good the language model is. In fact, catastrophic forgetting usually presents even if tasks are strictly non-overlapping (for example, when task identity is explicitly given during inference, referring to "task-incremental continual learning").

* The chosen experiment setting is not typical for catastrophic forgetting: catastrophic forgetting on language models is most clearly manifested in supervised task finetuning (e.g., on GLUE) and unsupervised domain finetuning (e.g., Codex, Minerva). Per my knowledge, there is no clear evidence of  catastrophic forgetting in instruction finetuning and RLHF yet (though forgetting is likely to exist). It would be better to choose a typical setting for evaluating the effectiveness of the proposed method.

* RLHF for alignment is not a good example of forgetting: the goal of alignment is to make the model refuse to answer when given a malicious instruction, rather than making the model forget how to follow harmful instructions. There is yet no generally effective method to make a model forget selected information (referring to "machine unlearning"). Catastrophic forgetting is usually recognized as a side-effect of finetuning and is direction-less (one cannot control exactly what is forgotten).

**Questions:**

* In section 2.5, what if the model is finetuned on a different set of discrete weights than $\mathcal{D}_{disk}$? Using a different set seems better matching real-world finetuning scenarios.
* It is observed that common LMs perform worse in languages other than English because of being primarily trained on English, so translation to another language is likely to reduce task performance (language gap). How does the reduction in performance because of forgetting compare to the reduction in performance because of the language gap? Would it be worth trading forgetting for the language gap?
* Could there be other kinds of mapping in conjugate prompting other than translation?

---

> ### Author Response · Authors · 2023-11-15
> **Response (1/2)**
>
> Thank you for the suggestions which have strengthened our paper! We see that your main concern is about the setup/assumptions. We appreciate your questions/concerns and hope that our additional experiments and clarification below convinces you that our setting is in fact quite broad, and does shed light on interesting aspects of possible performance regressions upon fine-tuning (which we broadly call catastrophic forgetting). Please let us know if you have any further questions or suggestions on how to present our findings appropriately.
>
>
>
> > The main assumption of the paper (Equation 7) is that the input space of different tasks overlaps, so the model has to do probabilistic inference on the task identity. … The ICL examples are purposefully designed to make it confusing by including two different instructions.
>
> > Per the author's theory, if the tasks' input spaces don't overlap, the model does not need to adjust the task probability g (as it would be already close to 1 for a good enough language model), and catastrophic forgetting would be much less if g does not need to be modified.
>
> 1. We would like to clarify that while the input space overlaps, the optimal mixture regression has close to zero “task ambiguity”---the optimal regressor can identify continuous vs discrete with near perfect accuracy even for a small number of exemplars. (Appendix C.5). However, the fine-tuned transformer is not optimal and performs imperfect task inference, which manifests as task ambiguity that is resolved via conjugate prompting.
> 2. We would also like to clarify that we prepend an instruction such as “Repeat the input” or “Capitalize every character” in our ICL experiments, so again, there is no inherent ambiguity. However, we do agree that this setup is somewhat artificial, and was intended as a sanity check that these phenomena do appear in large-scale real world LLMs as well.
> 3. We have added a new experiment that is in a more natural and commonly studied setting of forgetting. We consider Code LLaMa (LLaMa-2 fine-tuned on code) and see that there is a drop in performance on MNLI when compared to LLaMa-2, which can be recovered via conjugate prompting. We have more details below, as well as in Appendix D.
>
> > It can also be argued that when the input space of different tasks overlaps, the problem itself is ill-posed as there is not a single correct answer for inputs from the intersection. The failure of finetuned models thus may come from the ambiguity of the task itself rather than from catastrophic forgetting.
>
> Even if the problem is ill-posed, we note that the pre-trained model has a consistent preference for performing the task of interest on the input space in our three setups (ridge regression, ICL, harmful generation). Conjugate prompting hints that this drop comes from changing the model’s task inference rather than degrading the capability. Regardless of initial task ambiguity, the fine-tuned model faces even worse ambiguity from its changed task inference.
>
> > …catastrophic forgetting on language models is most clearly manifested in supervised task finetuning (e.g., on GLUE) and unsupervised domain finetuning (e.g., Codex, Minerva) … It would be better to choose a typical setting for evaluating the effectiveness of the proposed method.
>
> Thank you for this wonderful experiment idea! Following your suggestion, we consider Code-LLaMa vs LLaMa-2 evaluated on XNLI, a multi-lingual variant of the MNLI task from GLUE. We observe that Code-LLaMa displays catastrophic forgetting with an >8% accuracy drop on the English variant of this benchmark. However, with conjugate prompting into French, Spanish, and German, we see a less than 2% change in model performance, supporting our claim. In fact, we even see a slight increase in French and Spanish, possibly due to increased reasoning capabilities from code that are not suppressed by incorrect task inference [Ma et al, 2023](https://huggingface.co/papers/2309.16298). In this scenario, using Code LLaMa in Spanish gets higher performance than using LLaMa 2 in English. We discuss these new results in detail in Appendix D.

---

> ### Author Response · Authors · 2023-11-15
> **Response (2/2)**
>
> > RLHF for alignment is not a good example of forgetting: the goal of alignment is to make the model refuse to answer when given a malicious instruction, rather than making the model forget how to follow harmful instructions.
>
> We took a very broadly construed definition of forgetting to mean any “performance regression” (intentional or not) when fine-tuning a model and agree that catastrophic forgetting is perhaps overloaded. We have toned down our claims of RLHF being an intentional form of catastrophic forgetting and discussed how decreased answering can be studied under our task inference framework. Our results, as you mention, demonstrate that the model does not completely forget how to answer and retains the capability in a way that can be recovered by conjugate prompting.
>
> > In section 2.5, what if the model is finetuned on a different set of discrete weights than D_disc? Using a different set seems better matching real-world finetuning scenarios.
>
> We find that fine-tuning is commonly done for enhancing capabilities the model already has rather than endowing new capabilities since fine-tuning is usually not done on long tail data (i.e. instruction-tuning, alignment, domain fine-tuning, etc). Nonetheless, our experiments for $\alpha=0.1$ demonstrate how conjugate prompting helps even when the fine-tuning data is rarely present in the pretraining data.
>
>
> > How does the reduction in performance because of forgetting compare to the reduction in performance because of the language gap? Would it be worth trading forgetting for the language gap?
>
> We acknowledge this valid limitation of our method. However, for our ICL, RLHF, and code experiments, we find that the drop in base performance is small enough to potentially justify conjugate prompting over using the fine-tuned model in English. We also believe there may be other transformations where the base performance is not as big of an issue.
>
> > Could there be other kinds of mapping in conjugate prompting other than translation?
>
> Beyond languages, we test Pig-Latin and Leetspeak. For studying jailbreaking, [Wei et al, 2023](https://arxiv.org/abs/2307.02483) suggest transformations such as ROT-13 cipher, Morse Code, or synonyms with different connotations to push prompts out-of-distribution.
>
> —
>
> Thanks again for these detailed comments, we would love to know if there are any other ways we could strengthen our paper!

---

> ### Author Response · Authors · 2023-11-20
>
> We were wondering if there are any further questions or concerns we could clarify before the discussion period closes in two days. From your original review, it seems like the biggest concern was about the assumptions of our setting, especially as it relates to “catastrophic forgetting” in general. We appreciate your feedback and believe we addressed it in our response and additional real-world experiment. Please let us know if this addresses your concerns, and we’d be happy to discuss more in the remaining period of the rebuttal!

---

### Official Review · Reviewer_sicX · 2023-10-30

**Soundness:** 3 good
**Presentation:** 3 good
**Contribution:** 3 good
**Rating:** 8
**Confidence:** 3

**Summary:**

The authors propose to use translated prompts to study model behavior after fine-tuning outside of the fine-tuning distribution.

The first section describes a set of synthetic experiments where transformers are asked to learn linear functions, with a setup where the function is sampled from a continuous space of functions or from a fixed set of possibilities. In a single training scenario, Transformers learn both functions well, generalizing appropriately. The authors then train on a mixture of these two setups, showing that the tradeoff Transformer models achieve are quite suboptimal for the discrete setup. This is alleviated by more training on more examples, but the model approaches the Bayes error extremely slowly. The authors find that, unsurprisingly, models trained on the continuous setup and finetuned on the discrete setup rapidly become worse at the continuous setup. Next it is demonstrated that this forgetting is less prominent far from the fine-tuning description. The authors then describe a conceptual model of what they believe is going on—that the model is downweighting a prior of what problem is being solved, and snapping onto the fine-tuning task. Finally, the authors attempt to test this hypothesis by scaling in-context examples for the non-fine-tuned task out of the range of the fine-tuned task. This improves performance, providing evidence that pretrained abilities are not “lost” they are merely down-weighted in the prior of tasks the model presumes examples represent.

The authors go on to present they believe are sufficient conditions for accessing pretraining capabilities not selected-for in fine-tuning via prompting, which they term conjugate prompting. Next, the authors show that when fine-tuned for instruction following, models “forget” how to perform in-context learning, even when the model is prompted to perform ICL. This is interpreted as favoring the hypothesis that fine-tuning is largely changing how models estimate which task is being solved. This is used to motivate translation as a natural method of making prompts that are out-of-distribution for the fine-tuning set, but invertible back into it via translation back into English. The authors find that ICL capabilities can be recovered through translation. In the next subsection a similar experiment shows that models comply with instructions they would otherwise refuse when a similar translation strategy is used.

**Strengths:**

- The hypothesis is well-motivated, clearly expressed, and well operationalized: does finetuning get rid of tasks, or get rid of the model’s willingness to select certain tasks?
- While the synthetic experiments are somewhat artificial, they show a clear conceptual model of what kind of perturbations we may believe are going on in more realistic scenarios.
- The results are at least initially good evidence that the authors’ hypothesis is true in at least a limited way: models do not learn to up-weight fine-tuned tasks in other languages.

**Weaknesses:**

- More experiments should have been conducted on massively pretrained LLMs. The two experimental setups shown appear well considered and interesting, but many questions are raised such as: how much finetuning data is required to see conjugate prompting make a significant difference? Does this vary with model size significantly? Does this work the same across all tasks? A single paper cannot answer all of these questions, but they are not even acknowledged, and the lack of experiments addressing any of these variables makes the evidence somewhat thin.
- The synthetic experiments are interesting, but they do relatively little as evidence that this is what should be happening in large language models. I think they are useful, but given the fact that somewhat limited experiments were conducted on true LLMs, and claims are focused in that direction, it feels a little bit lopsided in terms of focus.
- Many descriptions are confusing. For instance in Table 2 it is unclear without significant interpretative discretion by the reader whether percentages represent refusal or lack of refusal. Or in section 2.6 it is stated that “For instance, in Table 1 it is unclear whether languages represent fine-tuning or prompting languages. From context, it becomes clear that this is for prompting, but only with a significant amount of digging. Clearer descriptions are necessary. This is not a minor issue—I had to reread most sections multiple times to understand what they were describing.

**Questions:**

- “As noted in prior work” on page 3, should really cite other work.
- You never identify IF as “Instruction Following”, while this is guessable from context, it took me a few minutes to figure this out and I recommend you introduce it properly.

---

> ### Author Response · Authors · 2023-11-15
> **Response**
>
> Thank you for the suggestions! We hope our responses and new experiments address any concerns, and we would love to know what else we can do to improve our paper!
>
> We agree with the reviewer that there is a lot of interesting follow-up work in validating our hypothesis across model scales, tasks, and conditions on the data distribution. To touch on some of these aspects, we run additional experiments for a larger model (Appendix C.6) and less/more pretraining data (Appendix C.7) in the synthetic setup. We also now acknowledge this as a limitation.
>
> We want to thank the reviewer for the suggestion to try less artificial tasks. In light of this, we validate conjugate prompting on XNLI, a multi-lingual variant of the MNLI task from GLUE. We observe that Code-LLaMa, a code-based fine-tune of LLaMA-2, displays catastrophic forgetting with an >8% accuracy drop on the English variant of this benchmark. However, with conjugate prompting into French, Spanish, and German, we see a less than 2% change in model performance, supporting our claim. In fact, we even see a slight increase in French and Spanish, possibly due to increased reasoning capabilities from code that are not suppressed by incorrect task inference [Ma et al, 2023](https://huggingface.co/papers/2309.16298). In this scenario, using Code LLaMa in Spanish gets higher performance than using LLaMa 2 in English. We discuss these new results in detail in Appendix D.
>
> We appreciate the reviewer’s suggestions to improve the clarity of our paper and have made the following modifications.
> - We have rewritten the captions of Table 1 and Table 2 for additional clarity on what is being measured and reported. If there were any other descriptions that would benefit from additional clarity, we would love to address them as well.
> - We have now cited the prior work alluded to on Page 3
> - We now identify IF as instruction following at the start of Section 4.1
>
> —
>
> Thanks again for these detailed comments, we would love to know if there are any other ways we could strengthen our paper!

---

> ### Author Response · Authors · 2023-11-20
>
> We were wondering if there are any further questions or concerns we could clarify before the discussion period closes in two days. We appreciate the feedback!

---

> > ### Comment · Reviewer_sicX · 2023-11-20
> >
> > Thank you for your clarifications! I appreciate the extra results, which do help validate things further. I will keep my score at an 8, pending further reviewer discussion.

---

### Official Review · Reviewer_X78o · 2023-11-01

**Soundness:** 3 good
**Presentation:** 3 good
**Contribution:** 3 good
**Rating:** 6
**Confidence:** 4

**Summary:**

This paper studies the effect of finetuning on pretrained transformer model's capabilities. The study is mostly empirically, involving both synthetic toy tasks on smaller transformers and more realistic tasks on medium size models.

The key proposition of the paper is
1) Finetuning harms pretrained capabilities as an interference effect, especially on inputs that are somewhat close but not the same as the finetuning data distribution.

2) It is possible to alleviate this effect by manually make the prompt more dissimilar to the finetuned distribution but keep their underlying sematics, for example by translating them to different numerical regimes or different languages.

**Strengths:**

The paper studies an important phenomenon - the effect of finetuning, in the setting of transformers / language models. These are main-stream practices in the field and will enjoy broad interests in the results.

The propositions of the paper are reasonable and are adequately supported by their empirical results.

The quantitative result on the impact of instruction tuning on ICL is particularly interesting.

The synthetic experiments are interesting and provide a toy case that can be understood better. In addition, the synthetic experiment also provides a simple framework that can be somewhat useful for real-world situations.

The paper is in general well-written.

**Weaknesses:**

The paper try to emphasize the difference between "forgetting" and "suppressing" of the pretrained capabilities. However this distinction might be spurious. For example, even though the English instruction tuned model may still be able to do an ICL task in Spanish, this does not mean that it did not "forget" how to do this task in English.

The method of proposed by the paper (re-formatting prompts) is a good simple trick that can be quickly applied to alleviate some forgetting. However it does not seem to attack the core of the problem, which is the fundamental over-specialization of the model onto the finetuning data. But as an analysis paper this may not be a critical weakness.

The synthetic task is too simple - different tasks there simply correspond to different linear weights, while pretraining vs finetuning are just different prior distribution of those weights. It might be beneficial to expand that - even in the regression setups - to e.g. higher order functions (quadratic, quartic etc), such that the transformer model's ability to generalize across a diverse set of tasks in pretraining, and how this could be changed by finetuning, can be tested better.

The related works section is somewhat thin. The paper might benefit from a more in-depth discussion of closely related works on forgetting in the field of transformer language models, particularly related to the effect of finetuning and ICL. For example: arxiv 2211.00635.

**Questions:**

Similar to the synthetic task, what happens if at the instruction-tuning stage, the fine tuning task is multi-task trained together with the pretraining task with different mixing weights, how would this change the behavior?

How do the behaviors observed scale with model size? Previous work (https://openreview.net/forum?id=GhVS8_yPeEa) has found that the forgetting becomes less of a problem when the model scales bigger, everything else being equal.

---

> ### Author Response · Authors · 2023-11-15
> **Response**
>
> Thank you for the suggestions! We hope our responses and new experiments address any concerns, and we would love to know what else we can do to improve our paper!
>
> > The paper try to emphasize the difference between "forgetting" and "suppressing" of the pretrained capabilities. However this distinction might be spurious. For example, even though the English instruction tuned model may still be able to do an ICL task in Spanish, this does not mean that it did not "forget" how to do this task in English.
>
> Whether the model solves the task prompted in English or Spanish, we say that it is using the same capability since the reasoning needed to perform ICL is independent of the language of the prompt. We consider “suppressing” to be when the model is capable of solving the task but doesn’t to separate out two possible worlds
> The first possibility is that, due to catastrophic forgetting, there’s no way to recover the lost ICL performance other than retraining the model to learn this capability again
> The second possibility is that the fine-tuned model can still perform ICL, but, due to catastrophic forgetting, won’t respond to the basic English input and needs to be prompted appropriately. We provide evidence for this scenario since conjugate prompting can partially recover the ICL performance.
> We would love to make this more clear in the paper and would appreciate any further suggestions on where we can best do so.
>
> > The synthetic task is too simple - different tasks there simply correspond to different linear weights, while pretraining vs finetuning are just different prior distribution of those weights. It might be beneficial to expand that - even in the regression setups - to e.g. higher order functions (quadratic, quartic etc), such that the transformer model's ability to generalize across a diverse set of tasks in pretraining, and how this could be changed by finetuning, can be tested better.
>
> We find that linear regression and task inference is a surprisingly difficult task for a transformer to implement, as prior work provides large constructions dedicated to implementing matrix inversion, multiple steps of gradient descent, or task inference ([Akyürek et al, 2022](https://arxiv.org/abs/2211.15661), [Bai et al, 2023](https://arxiv.org/abs/2306.04637)). Furthermore, the complex and unstudied task of mixture regression involves discrete regression (as discussed in Appendix B). We do agree with the reviewer that higher order functions could elucidate further interesting trends, and we think this would be interesting future work.
>
> > Similar to the synthetic task, what happens if at the instruction-tuning stage, the fine tuning task is multi-task trained together with the pretraining task with different mixing weights, how would this change the behavior?
>
> This is a great suggestion - we expect the multitasking to reduce forgetting but not completely mitigate it. As per our paper’s mental model, we would need to balance the pretraining and finetuning weights so that the task inference is not skewed one way or the other. Unfortunately, we did not have the compute to test this on real world LLMs, where we stuck to using publicly available fine-tuned models. However, we hope our work inspires such experiments and new methods of fine-tuning (and conjugate prompting) to mitigate performance regressions when fine-tuning.
>
> > How do the behaviors observed scale with model size? Previous work (https://openreview.net/forum?id=GhVS8_yPeEa) has found that the forgetting becomes less of a problem when the model scales bigger, everything else being equal.
>
> We introduce new experiments in Appendix C.6 to test what happens when we train on a larger model. We find that catastrophic forgetting still exists and conjugate prompting still helps. Interestingly, we find that they hold in similar magnitudes as well, showing how this problem is not resolved by simply scaling up. We also note that our real-world LLM experiments are on large scale models (such as ChatGPT3.5 and LLaMa). We encourage you to read our new section for more details!
>
> We will also write a more detailed related works section, discussing other forgetting works such as
> [Wang et al, 2023](https://arxiv.org/abs/2211.00635) which shows that forgetting in transformers can be mitigated through a two-stage fine-tuning scheme to prevent format specialization
> [Ramasesh et al, 2022](https://openreview.net/forum?id=GhVS8_yPeEa) which assesses forgetting and finds that models with more parameters or strong pretraining face less forgetting than their counterparts.
> [Luo et al, 2022](https://arxiv.org/abs/2308.08747) which asses forgetting in continual fine-tuning for large language models such as LLaMa, Alpaca, and BLOOMZ.
>
> —
>
> Thanks again for these detailed comments, we would love to know if there are any other ways we could strengthen our paper!

---

> ### Author Response · Authors · 2023-11-20
>
> We were wondering if there are any further questions or concerns we could clarify before the discussion period closes in two days. We appreciate the feedback, is there anything else we could add or discuss that might affect your score?

---

> ### Comment · Reviewer_X78o · 2023-11-21
>
> Would you like to include an explicit comparison of some metrics on "forgetting percentage rate", comparing the small model with the 2x model? The current presentation is hard for me to judge the scaling trend. Thanks!

---

> > ### Author Response · Authors · 2023-11-21
> >
> > Thanks for the suggestion! We've quantified the forgetting rates and success of conjugate prompting across model scale and alpha in Table 3, Appendix C.6, Page 21 of our updated paper. Though we discuss more details in the paper, we find that the forgetting drops are similar for both the 22.4 and 51.3M parameter models. We are currently running multiple training runs to provide error bars, which we hope to include in the next version of this paper. Is there anything else that would help your assessment?

---

> ### Comment · Reviewer_X78o · 2023-11-21
>
> Thanks for the new table! I think it helps but the question remains unclear - which is understandable given the short time frame.
>
> For example, we don't see clear gains from the larger model size before finetuning. If we conclude from this that model scaling has no effect on this task, then we are risking a wrong conclusion. More experiments / scale variations might be needed to see any signal one way or the other about forgetting too. Similarly, the author did not address several of my other questions due to limited time and resources. Again I do believe that the authors had put in their best efforts given the small amount of time they had.
>
> If I could give 6.5, I would.

---

> > ### Author Response · Authors · 2023-11-22
> >
> > We appreciate the score increase in spirit! We agree that the current experiment does not conclusively show the complete scale trends--we believe that we've trained for long enough on this task such that increasing the model size doesn't significantly improve performance. We'd like to highlight we still see considerable forgetting in the largest (regression and language) models we currently have and that forgetting persists in a simple setting even after training for long, motivating the importance of this problem. Thanks again for the feedback!

---

### Official Review · Reviewer_xPCE · 2023-11-01

**Soundness:** 4 excellent
**Presentation:** 4 excellent
**Contribution:** 3 good
**Rating:** 6
**Confidence:** 3

**Summary:**

A significant real-world problem with fine-tuned language models is the risk of the model forgetting how to perform tasks that it initially could handle. The authors found that fine-tuned models tend to perform more like their pre-trained counterparts on tasks that are far from the fine-tuning distribution. To address this, they introduce a method called Conjugate Prompting, which helps recover the pre-trained model's behavior by mitigating changes in implicit task inference. Five models and four non-English languages, with two additional transformations, demonstrate the effectiveness of this method.

**Strengths:**

1.	Well-written paper - clear and easy to read.
2.	The synthetic setup provides compelling evidence illustrating the trade-off resulting from fine-tuning and pre-training stages, specifically addressing the issue of catastrophic forgetting. The robustness of this analysis adds to the paper's credibility.
3.	Conjugate Prompting is simple but effective. The experiments conducted to validate this method are convincing, highlighting its practicality and potential benefits.

**Weaknesses:**

1.    In Section 2, the authors mentioned that the trade-off is less significant as the dataset size increases. Is this phenomenon consistent with the further expansion of the dataset, or is it merely an incidental occurrence specific to some certain smaller models?
2.    As the authors pointed out in the limitations section, this method would benefit from further validation and application in various domains and tasks involved.

**Questions:**

In Section 2, the authors fine-tuned a GPT-2 model and explored the phenomenon of trade-offs. However, it remains unclear whether this phenomenon is consistent across larger-scale language models or exhibits significant variations in performance improvements for different tasks, especially during the fine-tuning process. Are these differences in trade-offs related to the complexity or similarity of the tasks involved?

---

> ### Author Response · Authors · 2023-11-15
> **Response**
>
> Thank you for the suggestions! We hope our responses and new experiments address any concerns, and we would love to know what else we can do to improve our paper!
>
> > Is this phenomenon consistent with the further expansion of the dataset, or is it merely an incidental occurrence specific to some certain smaller models?
>
> > it remains unclear whether this phenomenon is consistent across larger-scale language models
>
> We believe our findings are consistent across data, model scale, and various tasks as evidenced by our experiments across linear regression and natural language (with 7B LLaMa and ChatGPT 3.5). To better demonstrate this, we provide extra experiments for larger models (Appendix C.6) and less/more pretraining data (Appendix C.7). In all three settings, we find that catastrophic forgetting is still a problem and conjugate prompting still improves performance on the continuous distribution. When we scale models, the trends stay roughly the same, though the base performance of the larger pretrained model is slightly stronger on the discrete distribution. We encourage you to look at these new sections for more details addressing these concerns!
>
> > Are these differences in trade-offs related to the complexity or similarity of the tasks involved?
>
> Though our hypothesis explains the behavior across all settings we consider, the exact trends will depend on confounding variables such as the complexity and similarity of tasks. We believe there is exciting future work in understanding the precise relationship between the fine-tuning data and the resulting model’s performance building off the viewpoint proposed in this work.
>
> > As the authors pointed out in the limitations section, this method would benefit from further validation and application in various domains and tasks involved.
>
> We want to thank the reviewer for the suggestion. In light of this, we validate conjugate prompting on XNLI, a multi-lingual variant of the MNLI task from GLUE. We observe that Code-LLaMa, a code-based fine-tune of LLaMA-2, displays catastrophic forgetting with an >8% accuracy drop on the English variant of this benchmark. However, with conjugate prompting into French, Spanish, and German, we see a less than 2% change in model performance, supporting our claim. In fact, we even see a slight increase in French and Spanish, possibly due to increased reasoning capabilities from code that are not suppressed by incorrect task inference [Ma et al, 2023](https://huggingface.co/papers/2309.16298). In this scenario, using Code LLaMa in Spanish gets higher performance than using LLaMa 2 in English. We discuss these new results in detail in Appendix D.
>
> —
>
> Thanks again for these detailed comments, we would love to know if there are any other ways we could strengthen our paper!

---

> ### Author Response · Authors · 2023-11-20
>
> We were wondering if there are any further questions or concerns we could clarify before the discussion period closes in two days. We appreciate the feedback, is there anything else we could add or discuss that might affect your score?

---

> > ### Comment · Reviewer_xPCE · 2023-11-23
> >
> > Thanks for the explanation!
> > I hold the original rating but lean towards accepting this paper.

---

### Author Response · Authors · 2023-11-15
**Summary of New Experiments**

We really appreciate the reviewers for giving us suggestions on how to improve our paper! Thanks to the comments, we’ve considered three new settings that provide further rigor to our claims. These experiments are currently available as appendix sections for clarity during rebuttals and will be appropriately incorporated into the main paper in our next version. We briefly summarize these below; we request you to read the corresponding appendix sections if you are looking for more detail.

1. Catastrophic forgetting for code fine-tuning: We consider one more forgetting scenario in real world LLMs where LLaMa-2 is fine-tuned on code to get Code LLaMa. Code Llama has worse performance than Llama-2 on MNLI in English. We then test the effect of conjugate prompting via language translation using the multi-lingual variants of this benchmark. We find very little forgetting in other languages, with Code LLaMa performing better in Spanish than in English, showing that conjugate prompting was able to reverse some of the forgetting induced by fine-tuning. We provide a more detailed discussion in Appendix D.
2. Testing effects for larger models: We recreate our experiments for a larger model with 51.3 million parameters (over double the parameter count of our original) to see if our results hold across the scale. We find that the overall trends hold: catastrophic forgetting occurs where performance on the continuous distribution drops as we fine-tune on the discrete distribution, and conjugate prompting can recover performance on the continuous distribution. The precise magnitude of these effects changes as we increase the model size, and we discuss this in more detail in Appendix C.6.
3. Testing effects for when scaling up data: In our paper, we train a model 5000 steps (i.e. 500k pretrained samples). We now also test what happens with fewer datapoints (200k) and more samples (1 million). We similarly find catastrophic forgetting occurs and this phenomenon does not seem to disappear with scaling up data , and conjugate prompting continues to help. We provide a more detailed discussion in Appendix C.7.


We hope our individual responses addressed any questions about our paper, and we would love to know what else we can do to improve the paper. Thanks again!

---

### Meta-Review · Area_Chair_L1Az · 2023-12-06

**Metareview:**

This paper presents a study of catastrophic forgetting in language models. Specifically, fine-tuning a model on a certain task distribution causes tasks that are slightly out of domain to be handled incorrectly; the paper proposes that this is due to shifting the distribution of "implicit task inference." This is shown in a setting of discrete and continuous regression.  A solution, conjugate prompting, is proposed to handle this, with experiments conducted in a multilingual instruction following setting, where translating instructions is a way of implementing conjugate prompting for LLMs.

The reviewers found this paper to study an interesting and important phenomenon. It nicely pairs experiments on synthetic data with a more realistic experiment. The paper is clear and well-written.

Several critiques of the paper are raised. X78o and sicX bring up the connection between the synthetic data and the real data setting; is the synthetic data so simple that this setting is not really a realistic model of what's happening in LLMs? I think it's useful on a conceptual level, but this is a fair weakness.

MyAe brings up a point about the paper's assumption about the input space of tasks overlapping, and whether this is realistic or not. I think there are several weaknesses of this review which are valid, namely the difference between the synthetic and real experiments and the nature of these tasks and their input spaces.

sicX offered a defense of the paper in discussion, which I will quote excerpts from from:

> However, the experiments do suggest that this kind of forgetting is happening, and the misnaming hardly makes this less worthy of study [...] The experiments show something is going on, the authors provide a hypothesis, suggest an algorithm for studying the effect they find, and provide evidence the algorithm works. [...] I think that while the synthetic experiments may well not represent the phenomena the authors are studying, they provide an interesting, falsifiable hypothesis for what the authors think is happening. Despite my own criticism of the synthetic experiments, I think it would be hard to argue that they definitively do not represent what is going on in the more realistic experiments, and so I feel this should not be a point against the authors.

I tend to agree with this point: although not perfect, this paper offers interesting evidence and a path forward for thought-provoking research on this topic.

Finally, sicX wants to see an analysis on a broader class of language models, which the authors addressed in the rebuttal.

**Justification For Why Not Higher Score:**

Reviews are mixed enough that a higher score is hard to justify

**Justification For Why Not Lower Score:**

See comments on reviewer discussion and the meta-review for why I ultimately think this should be accepted

---

### Decision · Program_Chairs · 2024-01-16

Accept (poster)